# Neural underpinning of a respiration-associated resting-state fMRI network

**Wenyu Tu[1,2], Nanyin Zhang[1,2,3]***

[1]The Neuroscience Graduate Program, The Huck Institutes of the Life Sciences, The Pennsylvania State University, University Park, United States; [2]Center for Neurotechnology in Mental Health Research, The Pennsylvania State University, University Park, United States; [3]Department of Biomedical Engineering, The Pennsylvania State University, University Park, United States

**Abstract** Respiration can induce motion and $CO_2$ fluctuation during resting-state fMRI (rsfMRI) scans, which will lead to non-neural artifacts in the rsfMRI signal. In the meantime, as a crucial physiologic process, respiration can directly drive neural activity change in the brain, and may thereby modulate the rsfMRI signal. Nonetheless, this potential neural component in the respiration–fMRI relationship is largely unexplored. To elucidate this issue, here we simultaneously recorded the electrophysiology, rsfMRI, and respiration signals in rats. Our data show that respiration is indeed associated with neural activity changes, evidenced by a phase-locking relationship between slow respiration variations and the gamma-band power of the electrophysiological signal recorded in the anterior cingulate cortex. Intriguingly, slow respiration variations are also linked to a characteristic rsfMRI network, which is mediated by gamma-band neural activity. In addition, this respiration-related brain network disappears when brain-wide neural activity is silenced at an isoelectrical state, while the respiration is maintained, further confirming the necessary role of neural activity in this network. Taken together, this study identifies a respiration-related brain network underpinned by neural activity, which represents a novel component in the respiration–rsfMRI relationship that is distinct from respiration-related rsfMRI artifacts. It opens a new avenue for investigating the interactions between respiration, neural activity, and resting-state brain networks in both healthy and diseased conditions.

*For correspondence:
nuz2@psu.edu

Competing interest: The authors declare that no competing interests exist.

## Editor's evaluation

This paper will be of interest to researchers studying control of respiration and also those developing functional magnetic resonance imaging methodology. The work provides insight into the relationship between brain activity (measured directly) and non-invasive functional magnetic resonance imaging measures. The authors find that the respiration signal is associated with the γ band in the cingulate cortex, and both the γ signal and respiration signal correlate with distributed neuronal networks across the brain. This contributes to our knowledge of the contribution of respiration on neuro and neuro-vascular signals during resting conditions.

## Introduction

Resting-state fMRI (rsfMRI), which measures spontaneous blood-oxygen-level-dependent (BOLD) signal, is a powerful tool for non-invasively investigating brain-wide functional connectivity (**Biswal et al., 1995**; **Fox and Raichle, 2007**; **Smith et al., 2009**). Due to its hemodynamic nature, the rsfMRI signal is susceptible to systemic physiological changes such as respiration and cardiac pulsations

**eLife digest** What does the brain do when we breathe? Humans and other animals with lungs depend on breathing to supply their cells with oxygen for energy production. Neurons in the brain are supplied oxygen through an intricate system of blood vessels. When active, neurons consume a lot of energy and require a steady supply of oxygen-rich blood. In fact, this relationship between blood vessels and activity of neurons in the brain is so tightly linked that to study neuron activity researchers and clinicians often use an approach called functional magnetic resonance imaging (fMRI) to analyze the flow of oxygenated blood in the brain.

This imaging technique allows scientists to map how active different parts of the brain are at any given time without the need for an invasive medical procedure. Unfortunately, fMRI results are affected by the cycles of inhalation and exhalation that take place while breathing, even when an individual is at rest. This is because the rate and depth of respiration can vary, resulting in the body moving unpredictably and in $CO_2$ levels fluctuating in the brain, which can lead to changes in fMRI signals that do not correlate with neuron activity. Such misleading measurements are called 'artifacts'. The assumption that these fMRI results do not represent real brain activity has meant that the effects of breathing on neuron activity in different parts of the brain is poorly understood.

To solve this issue, Tu and Zhang performed fMRI on rats and combined the results with measurements of the depth and rate of respiration, and with electrophysiology, an approach that allowed them to directly record the electrical properties of neurons. This allowed them to map out the network of neurons that become active in response to breathing.

The results show that breathing leads to a specific fMRI signal that can be distinguished from the artifacts introduced by fluctuating $CO_2$ levels and body movements. The signal correlates with the activity of neurons measured using electrophysiology and with breathing patterns, and it disappears when the electrical activity of neurons in the brain is suppressed, even if the rats are still breathing. This suggests that breathing affects brain activity that is independent of the previously described artifacts.

Future studies may focus on how the brain responds to breathing or how respiration itself is controlled by the brain, with the methods developed allowing researchers to explore regions of the brain that increase their activity while breathing. This clears the path towards investigating the neural mechanisms underlying therapies and exercises that focus on breathing.

(*Murphy et al., 2013*; *Liu, 2016*; *Caballero-Gaudes and Reynolds, 2017*; *Chen et al., 2020*), and these effects are usually treated as non-neuronal artifacts in rsfMRI data.

Respiration is a major physiological process that drives fluctuations in cerebral blood flow and oxygenation (*Liu, 2016*). Respiration can affect the BOLD signal (*Thomason et al., 2005*; *Abbott et al., 2005*; *Birn et al., 2006*) with two types of effects resulting from slow respiration variations and fast cyclic changes, respectively. Slowly varying changes (typically below 0.15 Hz) in breathing rate and depth, which can be quantified as respiration volume per time (RVT) (*Birn et al., 2006*), covary with arterial $CO_2$ (*Wise et al., 2004*; *Chang and Glover, 2009*; *Hoiland et al., 2016*; *Liu et al., 2017*) and affect the BOLD signal via vasodilatory effects and/or autonomic influences of the vessel tone (*Duyn et al., 2020*; *Picchioni et al., 2022*). On the other hand, faster respiratory cycles (i.e., periodic inspiration and expiration), accompanied by the corresponding chest and neck movement, can induce changes in the static magnetic field, which in turn lead to BOLD signal changes (*Glover et al., 2000*; *Raj et al., 2000*; *Windischberger et al., 2002*). Both effects can be effectively mitigated/removed by standard rsfMRI preprocessing methods (*Birn et al., 2006*; *Glover et al., 2000*).

In addition to the well-characterized respiration-related non-neural artifacts, there is evidence hinting a possible neural component in the respiration–rsfMRI interaction (*Yuan et al., 2013*; *Shams et al., 2021*). First, respiration can directly drive brain-wide neuronal oscillations (*Tort et al., 2018a*; *Kay et al., 2009*) not only in the olfactory bulb (*Adrian, 1942*) and piriform cortex (*Fontanini et al., 2003*)—brain regions directly related to breathing—but also in the medial prefrontal cortex (mPFC), somatosensory cortex, and hippocampus, and this effect has been consistently found across species (*Ito et al., 2014*; *Yanovsky et al., 2014*; *Biskamp et al., 2017*; *Zelano et al., 2016*). In addition, respiration changes can be associated with arousal and/or emotion-related brain state changes, which

covary with cortical activity (*Yackle et al., 2017*; *Shea, 1996*; *Homma and Masaoka, 2008*; *Folsch-weiller and Sauer, 2021*). Therefore, in addition to the artifactual effects aforementioned, respiration may affect the rsfMRI signal by directly modulating the neural activity. However, this potential neural component in the respiration–fMRI relationship is largely unexplored.

To gain a comprehensive understanding of the relationships between respiration, neuronal activity, and rsfMRI signal, here we simultaneously acquired rsfMRI, electrophysiology, and respiration data in anesthetized rats. Anesthesia was used to ensure our results are not confounded by the animal's motion, which affects all three signals. Based on these measures, an RVT-correlated rsfMRI network was identified. Importantly, regressing out gamma activity or silencing neural activity across the brain disrupted this respiration-related network, suggesting that this respiration–rsfMRI relationship is mediated by neural activity.

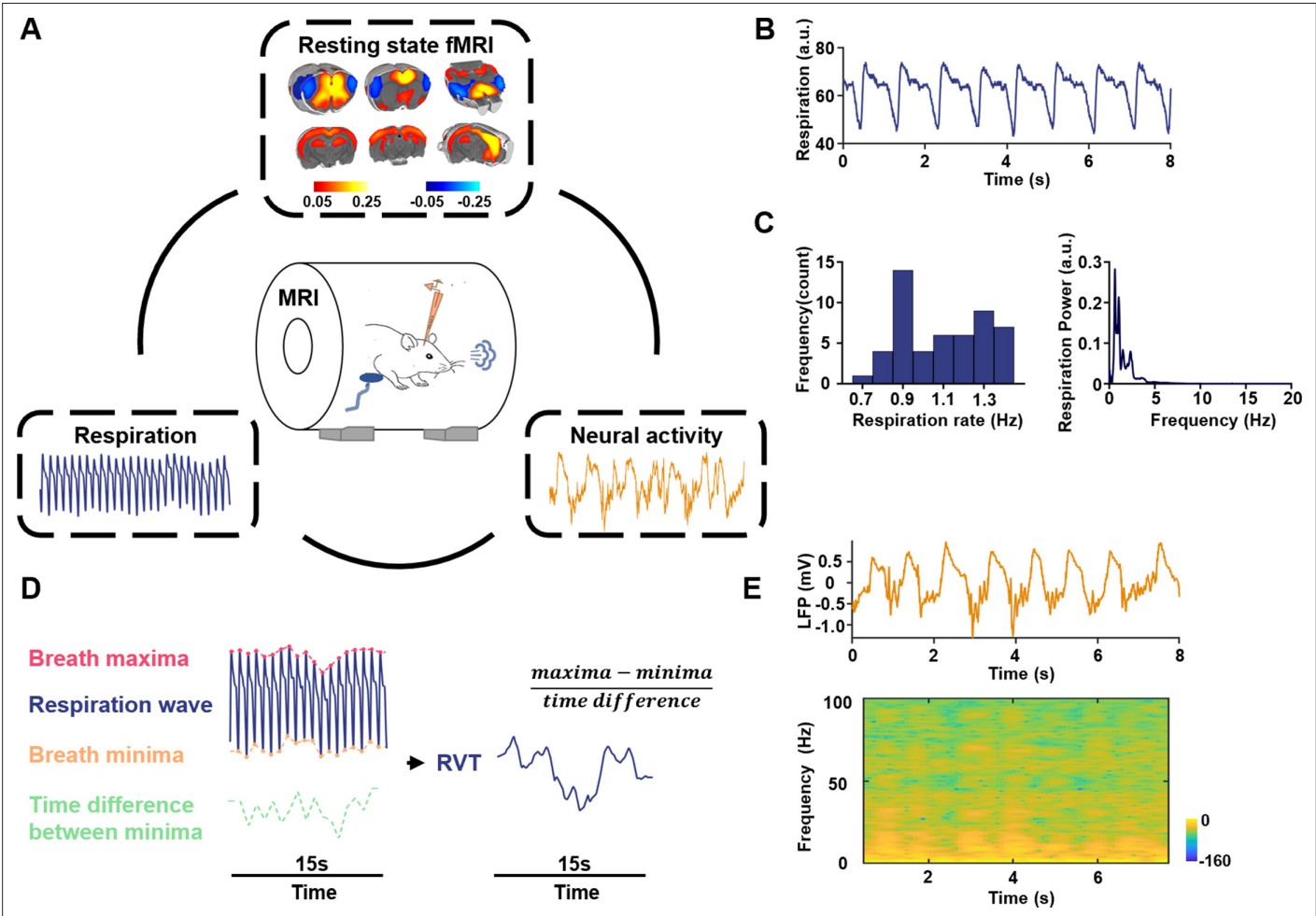

**Figure 1.** Simultaneous recordings of fMRI, electrophysiology and respiration signals in rats. (**A**) Experimental design—simultaneous measurement of the resting-state fMRI (rsfMRI), electrophysiology, and respirational signals. Top: anterior cingulate cortex (ACC) seedmap; bottom left: respiration signal; bottom right: local field potential (LFP). (**B**) Exemplar respiration signal waveform. (**C**) Left: distribution of the respiration rate across all scans; right: power of the respiration signal averaged across all scans. (**D**) Computing respiration volume per time (RVT) from the respiration waveform. (**E**) Exemplar denoised LFP signal. Top: LFP time series; bottom: LFP power spectrogram.

The online version of this article includes the following figure supplement(s) for figure 1:

**Figure supplement 1.** Representative image confirming the electrode location in the anterior cingulate cortex.

**Figure supplement 2.** Removal of MRI artifacts from the electrophysiology signal.

# Results

To determine the potential role of neural activity in the respiration–rsfMRI relationship, we simultaneously recorded the electrophysiology and respiration signals along with rsfMRI data in rats (*Figure 1A*). The respiration signal was recorded by a respiration sensor placed under the animal's chest (*Figure 1A*). Representative raw respiration signal, the respiration rate distribution across all scans, and the averaged respiration power are shown in *Figure 1B, C*. Slow respiration variations are quantified by RVT, calculated as the difference of consecutive peaks of inspiration and expiration divided by the time interval between the two adjacent signal maxima (or minima) (*Figure 1D*; *Birn et al., 2006*). The electrophysiology signal was recorded using an MR-compatible electrode implanted in the right side of the anterior cingulate cortex (ACC). The ACC was selected given its critical role in respiratory modulation, evidenced by early studies showing that ACC was activated during breathlessness (*Liotti et al., 2001*; *Evans et al., 2002*). More recent research further demonstrates the role of ACC in respiratory control (*Evans et al., 2009*; *Holton et al., 2021*; *Tort et al., 2018b*). For example, electrophysiology recordings in humans showed that ACC exhibited different neural responses to separate respirational tasks including breath-holding, voluntary deep breathing, and hypercapnia (*Holton et al., 2021*). In addition, the ACC is a key region in the rodent default-mode

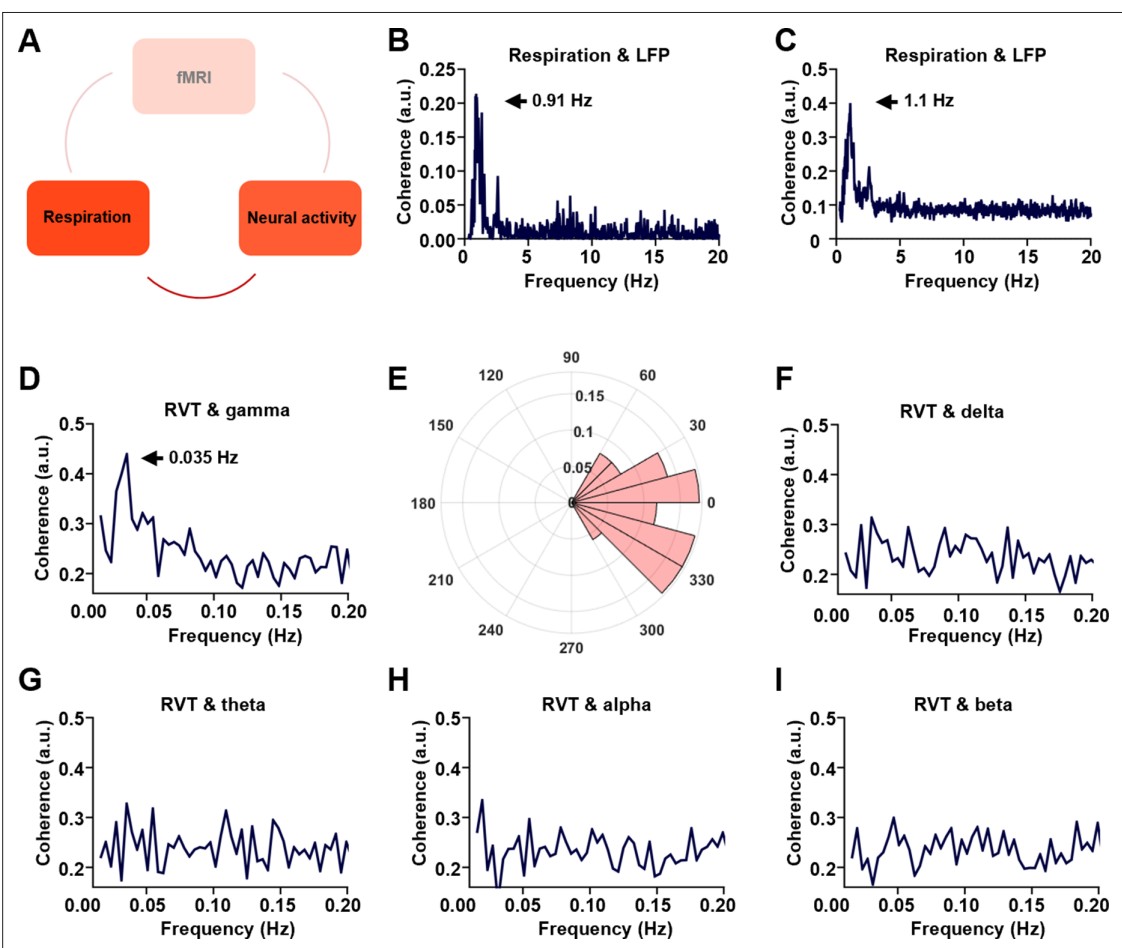

**Figure 2.** Phase-locking relationship between slow respiration variations and neural activity. (**A**) The relationship between respiration and neural activity. (**B**) Respiration–local field potential (LFP) coherence from one sample scan. (**C**) Respiration–LFP coherence averaged across all scans. (**D**) Coherence between respiration volume per time (RVT) and gamma-band power (40–100 Hz), with the peak at 0.035 Hz. (**E**) Phase lag between RVT and gamma-band power. In contrast, no obvious coherence is observed between RVT and delta-band power (1–4 Hz, **F**), theta-band power (4–7 Hz, **G**), alpha-band power (7–13 Hz, **H**), or beta-band power (13–30 Hz, **I**).

The online version of this article includes the following figure supplement(s) for figure 2:

**Figure supplement 1.** Electrophysiology results obtained using the differential subtraction method.

**Figure supplement 2.** Respiration volume per time (RVT)–gamma power coherence in individual animals in the lightly sedated state.

network (DMN) (*Lu et al., 2012*; *Tu et al., 2021a*), which has been linked to respiration-related fMRI signal changes (*Birn et al., 2006*). The location of the electrode was confirmed by T2-weighted structural images (*Figure 1—figure supplement 1*). MRI artifacts in the electrophysiology signal were removed using a template regression method (*Logothetis et al., 2001*; *Pan et al., 2011*) (see Materials and methods and *Figure 1—figure supplement 2*), and local field potential (LFP) and spectrogram were obtained using denoised electrophysiologic data (*Figure 1E*).

## Gamma-band neural oscillations are, respectively, associated with respiration and rsfMRI signals

We first asked whether the respiration is linked to neural activity changes (*Figure 2A*) in lightly sedated animals (combined low-dose dexmedetomidine and isoflurane, see Methods and materials). Prominent coherence between the respiration signal and LFP is observed with the dominant peak at ~1.1 Hz (*Figure 2B, C*), in line with the respiration frequency in animals (*Figure 1C*). This result is consistent with the finding of respiration-entrained LFP oscillations previously reported (*Tort et al., 2018a*; *Kay et al., 2009*).

To further dissect the respiration–LFP relationship, the LFP was separated into five conventionally defined frequency bands including gamma (40–100 Hz), delta (1–4 Hz), theta (4–7 Hz), alpha (7–13 Hz), and beta band (13–30 Hz) (*Lu et al., 2016*; *Zhang et al., 2020*). Gamma-band power displays significant coherence with the RVT at ~0.035 Hz with a zero phase lag (*Figure 2D, E*, phase [mean ± ste] = −0.08 ± 0.5 pi, with the range of [−pi, pi]), suggesting a phase-locking relationship between the two measures. In contrast, this coherence is not observed in any other frequency bands (*Figure 2F–I*). These results remain the same when the LFP data were analyzed using the local subtraction method (*Figure 2—figure supplement 1A–C*), ruling out the potential artifact resulting from the volume conduction of signals between the cortex and olfactory bulb (*Parabucki and Lampl, 2017*). The RVT–gamma power coherence in individual lightly sedated animals is shown in *Figure 2—figure supplement 2*. Taken together, our data demonstrate that respiration is associated with gamma-band neural oscillations in the ACC.

We next examined the relationship between the gamma-band LFP and rsfMRI signals (*Figure 3A*; *Winder et al., 2017*). The ACC gamma power was first convolved with a hemodynamic response function (HRF, defined by a single gamma probability distribution function ($a = 3$, $b = 0.8$) *Liang et al., 2017*, *Figure 3B*), which was then voxel-wise correlated to brain-wide rsfMRI signals. The $r$ value between the ACC rsfMRI signal and HRF-convolved gamma power was 0.086 (1200 time points, p = 0.003). Our data show that the gamma power-derived rsfMRI correlation map (*Figure 3C*) is highly consistent with the ACC resting-state functional connectivity (RSFC) seedmap with the right ACC (i.e., the electrode implanted side) defined as the seed (*Figure 3D*), evidenced by a strong voxel-to-voxel spatial correlation between these two maps (*Figure 3E*, R = 0.775, p < $10^{−15}$). Again, the same gamma power-derived correlation pattern is observed when the LFP data were analyzed using the local subtraction method (*Figure 2—figure supplement 1D*). The gamma power-derived rsfMRI correlation maps in individual animals are shown in *Figure 3—figure supplement 1*. These data demonstrate that the gamma-band LFP is tightly linked to the rsfMRI signal.

## Respiration is associated with a characteristic rsfMRI network, mediated by gamma-band neural activity

Given that the gamma-band power is, respectively, associated with the RVT and rsfMRI signals, we specifically asked how RVT is related to the rsfMRI signal in lightly sedated animals by calculating voxel-wise correlations between the RVT and rsfMRI signals (*Figure 4A*). This analysis generates a respiration-related rsfMRI network (*Figure 4B*), involving key brain regions controlling respiration such as the piriform cortex. It is known that the piriform cortex receives inputs from the olfactory bulb, and can be directly activated when the animal breathes. In addition, this network includes regions involved in the rodent DMN such as the ACC, mPFC, orbital, retrosplenial, and primary somatosensory cortices, as well as the hippocampus (*Lu et al., 2012*; *Figure 4C*, one-sample *t*-test, p < 0.05, false discovery rate [FDR] corrected). The resemblance of this respiration-related network and DMN well agrees with the human literature that the physiological effects on rsfMRI data are colocalized with the DMN (*Birn et al., 2006*). The respiration-related rsfMRI network in individual animals is shown *Figure 4—figure*

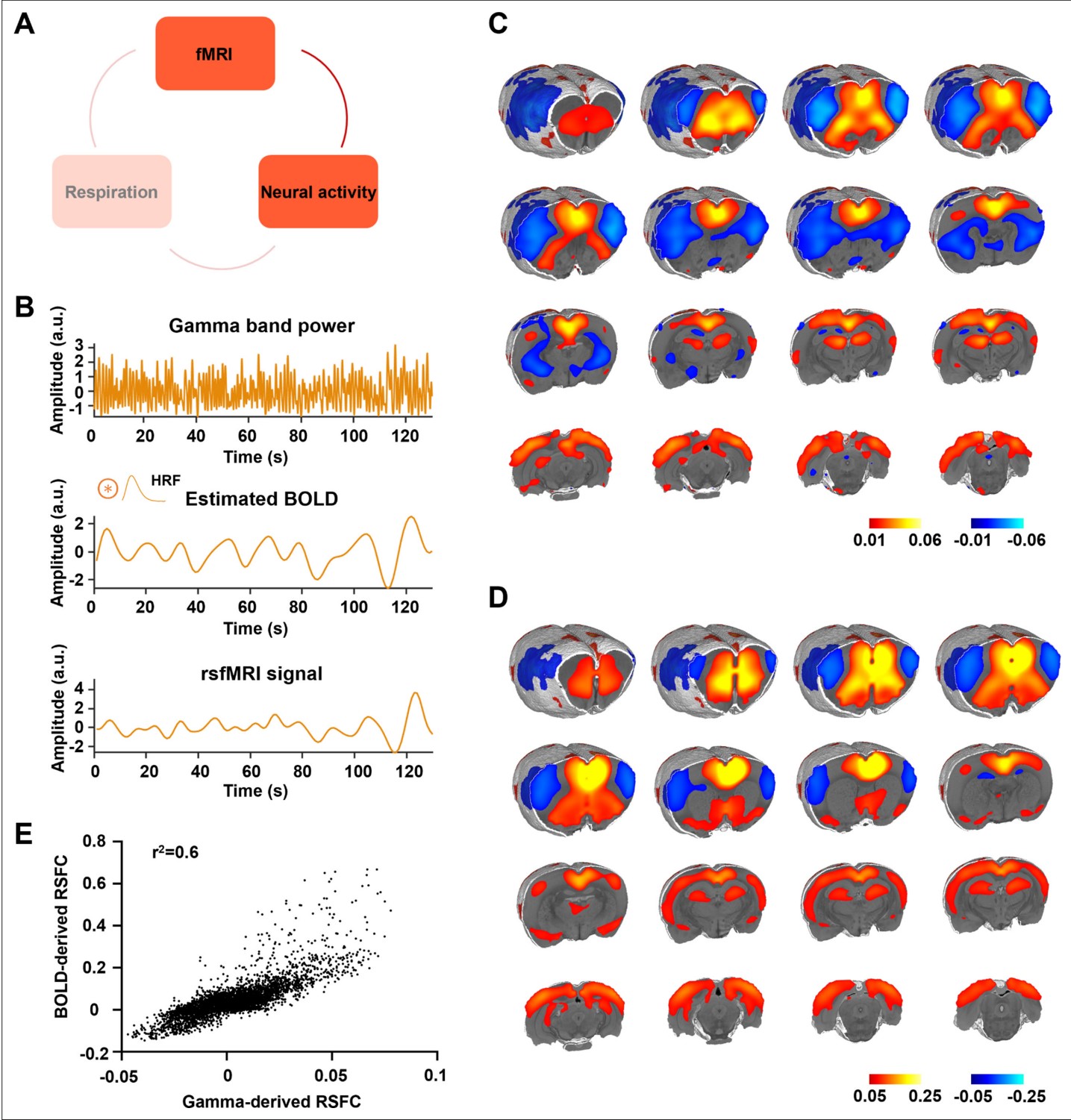

**Figure 3.** Gamma power is associated with the resting-state fMRI (rsfMRI) signal. (**A**) The relationship between neural activity and rsfMRI signal. (**B**) Top: exemplar gamma-band power in the anterior cingulate cortex (ACC); middle: estimated blood-oxygen-level-dependent (BOLD) signal by convolving the gamma-band power with the hemodynamic response function (HRF); bottom: measured BOLD signal from the same brain region. (**C**) Gamma power-derived correlation map. (**D**) Seedmap of the right ACC. (**E**) Voxel-to-voxel spatial correlation between (**C**) and (**D**).

The online version of this article includes the following figure supplement(s) for figure 3:

**Figure supplement 1.** Gamma power–resting-state fMRI (rsfMRI) correlation maps in individual animals in the lightly sedated state.

**Figure supplement 2.** Difference of RSFC maps before and after respiration volume per time (RVT) regression.

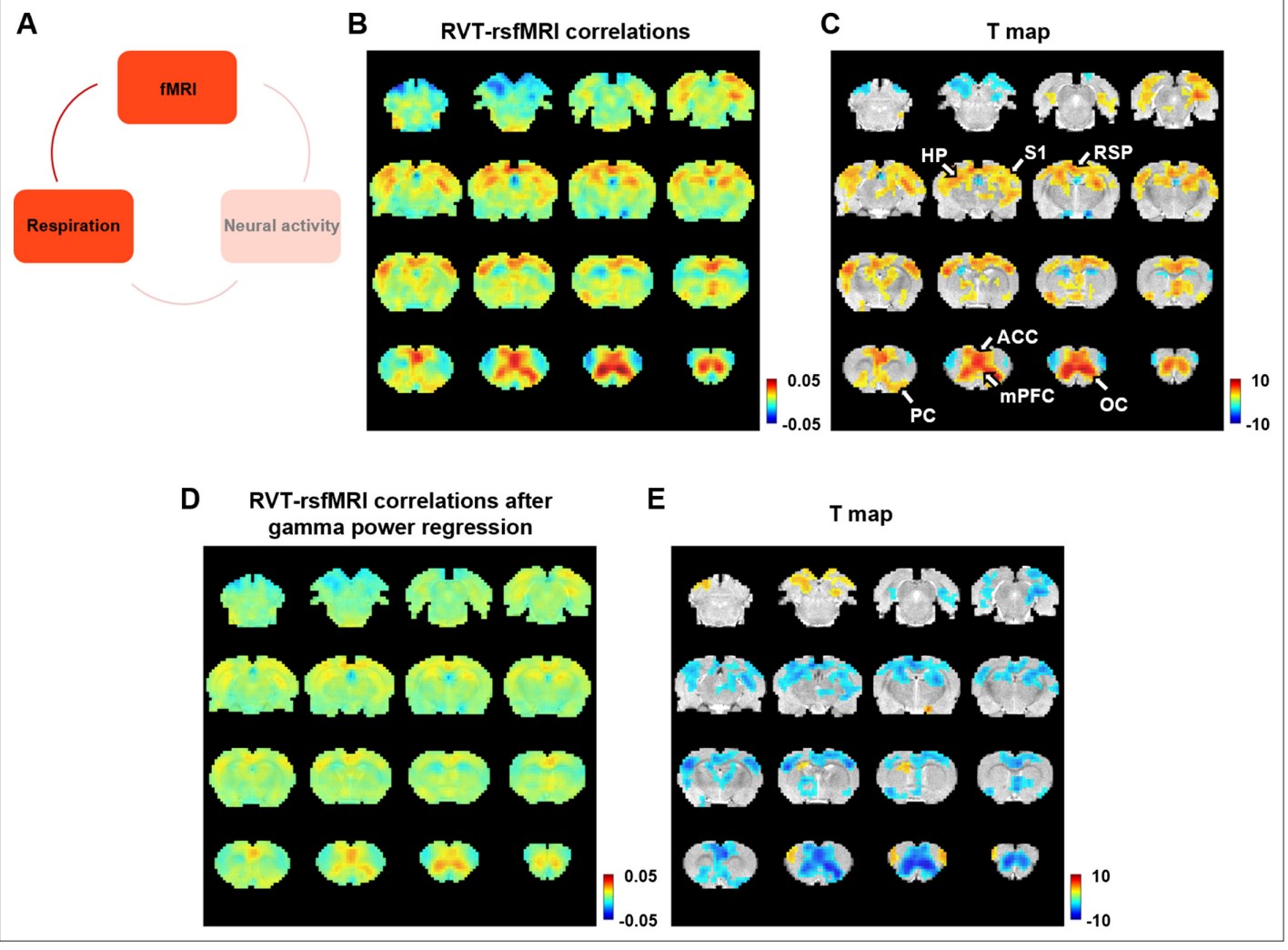

**Figure 4.** Correlation between slow variations of respiration and resting-state fMRI (rsfMRI) signal. (**A**) The relationship between respiration and rsfMRI signals. (**B, C**) Voxel-wise correlations between the respiration volume per time (RVT) and rsfMRI signals. (**B**) Unthresholded correlation map averaged across scans. (**C**) Thresholded *T*-value map (one-sample *t*-test, p < 0.05, false discovery rate [FDR] corrected). Brain regions displaying significant RVT–rsfMRI correlations include the anterior cingulate cortex (ACC), orbital cortex (OC), medial prefrontal cortex (mPFC), piriform cortex (PC), hippocampus (HP), retrosplenial cortex (RSP), and primary somatosensory cortex (S1). (**D**) Voxel-wise correlations between the RVT and rsfMRI signals after the gamma-band power are regressed out from both signals. (**E**) Difference of correlation maps before and after gamma power regression (paired *t*-test, p < 0.05, FDR corrected).

The online version of this article includes the following figure supplement(s) for figure 4:

**Figure supplement 1.** Respiration volume per time (RVT)–resting-state fMRI (rsfMRI) correlation maps in individual animals in the lightly sedated state.

**Figure supplement 2.** Voxel-wise correlations between the resting-state fMRI (rsfMRI) signal and RETROICOR regressor with the regression of the white matter and ventricle signals in (**A**) light sedation and (**B**) isoelectric state.

**Figure supplement 3.** Voxel-wise correlations between the estimated time course of respirational artifacts and resting-state fMRI (rsfMRI) signals.

*supplement 1*. Taken together, slow respiration variations exhibit a characteristic correlation pattern with brain-wide rsfMRI signals, representing a respiration-related rsfMRI network.

To test whether this respiration-related rsfMRI network is mediated by gamma-band activity, the gamma power was, respectively, regressed out from the RVT and voxel-wise rsfMRI signals, and the same correlational analysis was repeated on the residual signals. In this case, the respiration-related network is diminished (*Figure 4D*). This result is confirmed by the contrast of the correlation maps before (*Figure 4B*) and after (*Figure 4D*) gamma power regression, with essentially all brain regions involved in the respiration-related network exhibiting reduced RVT–rsfMRI correlation after

the gamma power is regressed out (*Figure 4E*, paired *t*-test, p < 0.05, FDR corrected). These results demonstrate that the respiration-related rsfMRI network is mediated by gamma-band neural activity.

## The respiration-related rsfMRI network is absent at the isoelectric state

To further confirm the necessary role of neural activity in the respiration-related rsfMRI network, we experimentally silenced the neural activity in the whole brain by inducing an isoelectric brain state using high-dose sodium pentobarbital, while maintained the respiration in the rat (*Figure 5A*; *Du et al., 2008*). At the isoelectric state, the LFP amplitude and power are close to zero, in remarkable contrast to the electrophysiology data recorded before the drug infusion (*Figure 5B*). In addition, rsfMRI data recorded at the isoelectric state exhibit a flat, noise-like power distribution, unlike the characteristic 1/f pattern during light sedation (*Figure 5I*; *Zhang et al., 2021*). Furthermore, there is no meaningful ACC RSFC in the ACC seedmap at the isoelectric state (*Figure 5H*), which is distinct from the ACC seedmap during light sedation (*Figure 3D*). The brain-wide ROI-based RSFC matrix also reveals a global suppression of RSFC at the isoelectric state (*Figure 5K*), compared to the RSFC matrix observed during light sedation. These data collectively confirm that brain-wide neural activity is silenced in the isoelectric state.

To examine whether the temporal variation of respiration in the isoelectric state and light sedation is consistent, we calculated the respiration rate (*Figure 5D*) as well as the standard deviation (SD) and power spectrum of RVT across all scans. We did not observe any significant difference in the SD of the respiration rate (*Figure 5E*, two-sample *t*-test, t = −1.0, p = 0.32) or the SD of RVT (*Figure 5F*, two-sample *t*-test, t = −0.87, p = 0.39) between the light sedation and isoelectric states, suggesting that the temporal variance of respiration is similar between the two states. Interestingly, we observed a characteristic 1/f pattern in the RVT power during light sedation, but this pattern was not present at the isoelectric state (*Figure 5G*). This difference is consistent with the ACC BOLD signal power spectra in the light sedation and isoelectric states (*Figure 5I*).

After confirming the global neural silencing effect, we calculated voxel-wise correlations between the RVT and rsfMRI signals in the isoelectric state. The respiration-related network observed during light sedation is absent when brain-wide neural activity is silenced (*Figure 5J*), despite similar breathing patterns (*Figure 5C–F*) at both conditions. The RVT–rsfMRI correlation maps in individual animals in the isoelectric state are shown in *Figure 5—figure supplement 1*. These data demonstrate that the neural activity plays a necessary role in the respiration-related rsfMRI network, corroborating the notion that the respiration-related rsfMR network we observed is mediated by neural activity.

## Respiration-related brain network is distinct from respiration-related rsfMRI artifacts

To confirm the respiration-related neural network we observed is different from respiration-related rsfMRI artifacts previous reported, we separately obtained the artifactual patterns of fast (i.e., cyclic inspiration and expiration) and slow respiration signals in the rsfMRI data.

The fast variations of respiration can cause aliased respiratory artifacts in the fMRI signal by inducing B0 offsets (*Pais-Roldán et al., 2018*; *Zhao et al., 2000*). To determine this effect, we calculated voxel-wise correlations between the rsfMRI signal and the respiration regressor estimated by RETROICOR (see Materials and methods for details) (*Glover et al., 2000*). Unlike the respiration-related neural network described above, we did not observe any appreciable correlations between RETROICOR and the rsfMRI signals across the brain in either the light sedation or isoelectric state (*Figure 4—figure supplement 2*). This result remains the same with and without regressing out the signals of the white matter and ventricles in rsfMRI preprocessing, likely due to minimal motion of animals in both anesthetized states.

To assess the artifactual effects resulting from slow respiration signals on rsfMRI data, we adopted the standard analysis method by convolving the RVT with a respiration response function (RRF) determined by the difference of two gamma variate functions (see Materials and methods, *Figure 4—figure supplement 3A*; *Birn et al., 2008*), and then voxel-wise correlated it to the rsfMRI signal (*Figure 4—figure supplement 3*). Note that this calculation is different from the method we used to identify respiration-related neural network, which directly correlated the RVT with the rsfMRI signal. Our data show that the slow respiration-related rsfMRI artifact is distinct from the respiration-related neural network. First, the slow respiration-related rsfMRI artifact is dominantly located around the

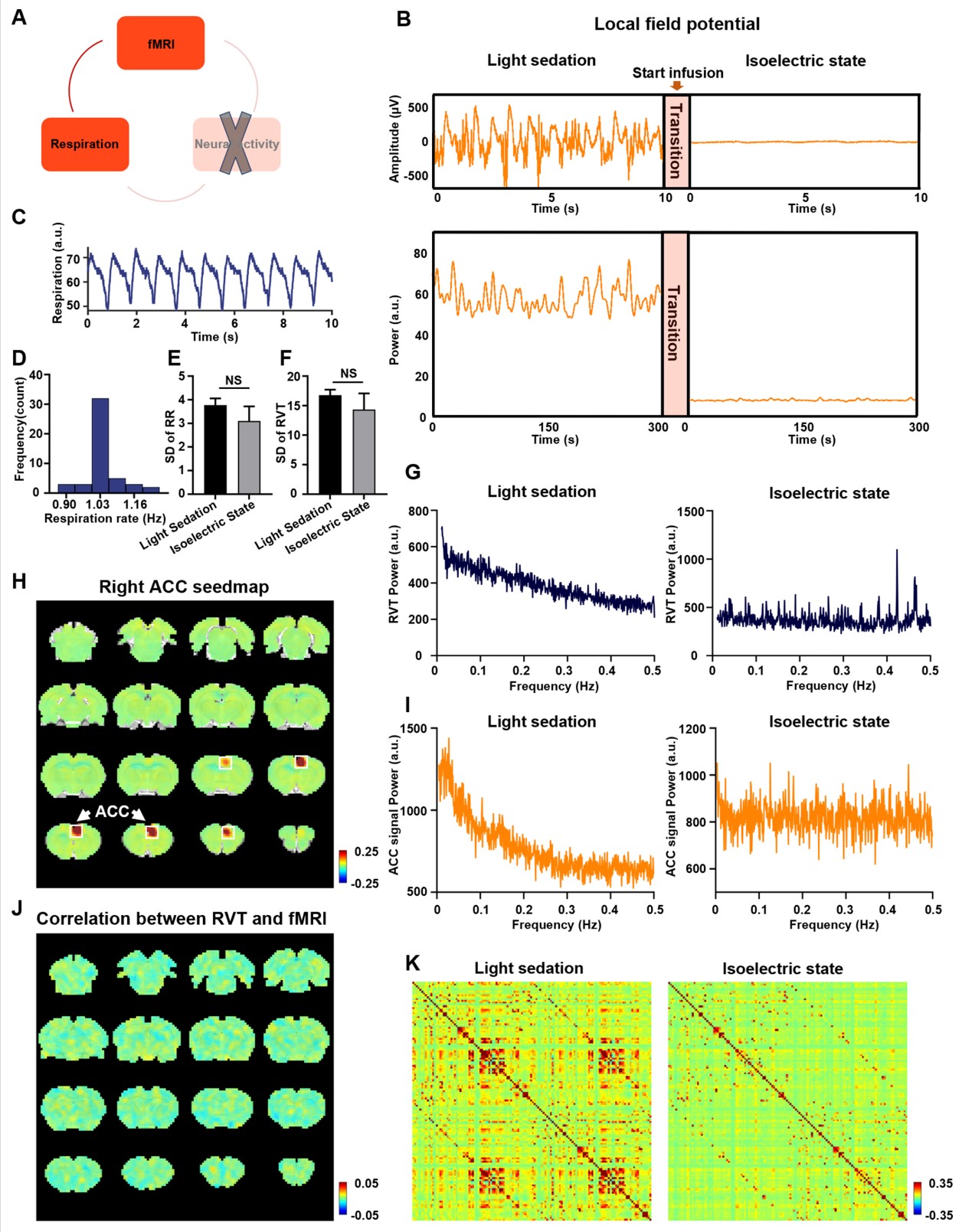

**Figure 5.** Respiration–resting-state fMRI (rsfMRI) relationship at the isoelectric state. (**A**) Determining the relationship between slow respiration variations and the rsfMRI signal after silencing the brain-wide neural activity. (**B**) Silencing neural activity at the isoelectric state induced by sodium pentobarbital. Top: LFP amplitude; bottom: LFP power; arrow: infusion of sodium pentobarbital. (**C**) Respiration signal at the isoelectric state. (**D**) Distribution of respiration rate (RR) across scans. (**E**) Variance of RR, quantified by the standard deviation (SD) of RR in the light sedation and isoelectric

*Figure 5 continued on next page*

Figure 5 continued

states. (**F**) Variance of respiration volume per time (RVT), quantified by the SD of RVT in the light sedation and isoelectric states. There is no significant difference in the SD of the RR (**E**, two-sample *t*-test, $t = -1.0$, $p = 0.32$) or the SD of RVT (**F**, two-sample *t*-test, $t = -0.87$, $p = 0.39$) between the light sedation and isoelectric states. (**G**) Power spectrum of RVT during (left) light sedation and (right) the isoelectric state. (**H**) Seedmap of the right anterior cingulate cortex (ACC) at the isoelectric state. (**I**) Power spectra of the blood-oxygen-level-dependent (BOLD) signal during (left) light sedation and (right) the isoelectric state. (**J**) Voxel-wise correlations between RVT and the rsfMRI signal at the isoelectric state. (**K**) Brain-wide ROI-based RSFC matrices at (left) light sedation and (right) the isoelectric state (132 ROIs in total).

The online version of this article includes the following figure supplement(s) for figure 5:

**Figure supplement 1.** Respiration volume per time (RVT)–resting-state fMRI (rsfMRI) correlation maps in individual animals in the isoelectric state.

ventricles and large veins (*Figure 4—figure supplement 3B*), whereas the respiration-related neural network is mostly located at the gray matter with a distinct spatial pattern (*Figure 4B, C*). In addition, the slow respiration-related rsfMRI artifact can be reduced by regressing out the signals in the white matter and ventricles (*Figure 4—figure supplement 3C*), but the respiration-induced neural network persists after the regression. Taken together, these data indicate that the respiration-related neural network we observed is not a non-neural physiological artifact but originates from neural activity.

## Discussion

In this study, we discovered a slow respiration variation-related functional network in rodents. Importantly, we confirmed the neural underpinning of this network, indicated by the phase-locking relationship between the RVT and gamma-band power in the ACC—a node in this network. Regressing out the gamma-band power disrupted the respiration network, suggesting a mediating role of neural activity in this network. To further validate this finding, we experimentally silenced the neural activity across the brain but maintained the respiration in the animal. In this condition, we again failed to observe the respiration-related brain network, confirming the neural underpinning of the network. Lastly, we showed that the respiration-related neural network is distinct from respiration-related rsfMRI artifacts. Overall, our study maps brain-wide neural responses related to respiration. These data provide new insight into understanding the neural activity-mediated respiratory effects on resting-state functional networks. Importantly, this breathing-related brain network might be altered in brain disorders, and thus our findings might potentially provide important clinical value.

### The respiration network is very likely linked to respiration-entrained brain-wide neural oscillations

Respiration-induced neuronal oscillations have been consistently observed across multiple species including rats, mice, and humans (*Yanovsky et al., 2014*; *Zelano et al., 2016*; *Lockmann et al., 2016*). This phenomenon also seems to be robust in various physiological and behavioral states including freely moving awake and immobile awake conditions, as well as anesthetized states (*Fontanini et al., 2003*; *Ito et al., 2014*; *Zhong et al., 2017*). Intriguingly, previous studies found that respiration specifically modulated the amplitude of gamma activity in the mPFC (*Zhong et al., 2017*; *Rojas-Líbano and Kay, 2008*; *Fontanini and Bower, 2005*), consistent with our finding of band-specific coherence between the gamma power in the ACC and RVT. Notably, gamma power is generally believed to relate to the BOLD signal (*Logothetis et al., 2001*; *Winder et al., 2017*), which is also demonstrated in our study (*Figure 3*). These results suggest that respiration is associated with gamma oscillations, which is necessary for BOLD signal changes in the respiration network. This notion is further supported by our data that regressing out gamma-band LFP signals tremendously reduced the correlation between RVT and fMRI (*Figure 4D,E*), although it is still possible that if there are non-neural hemodynamic effects of respiration on the rsfMRI signal that occur at zero lag (i.e., a non-neural hemodynamic factor that are synchronized with gamma LFP), regressing out gamma LFP could also reduce these effects.

In addition to the mPFC, previous work has shown that respiration-generated neuronal oscillations are phase locked to the breath rhythm across distributed brain regions including the olfactory bulb, primary piriform cortex, hippocampus, and somatosensory barrel cortex (*Fontanini et al., 2003*; *Ito et al., 2014*; *Yanovsky et al., 2014*; *Biskamp et al., 2017*; *Herrero et al., 2018*). Those regions are highly consistent with brain regions in the respiration network we observed. Furthermore, the piriform cortex is anatomically connected to the mPFC, ACC, and orbital cortex, all of which are parts of the

respiration network in our study (*Schmahmann, 2009*; *Illig, 2005*; *García-Cabezas and Barbas, 2014*; *Carmichael et al., 1994*). Taken together, the spatial pattern of the respiration network reported here well agrees with brain regions exhibiting neural oscillation changes driven by respiration.

Previous work also demonstrates that respiration-induced neuronal oscillations do not result from the movement of the muscle or electrode. For instance, the oscillations are different in the laminar amplitude along the hippocampus, with the maximal amplitude in the dentate gyrus (DG) (*Yanovsky et al., 2014*). This finding well agrees with our data, displaying more prominent involvement of the DG than other parts of the hippocampus in the respiration network. In addition, our data show that only the gamma-band power, not other LFP bands, is coherent with slow variations of respiration, whereas the movement of electrode or muscle would lead to increased coherence across multiple frequency bands. Taken together, the respiration-related functional network we observed is very likely linked to the respiration-entrained gamma-band oscillations in regions involved in the network.

Notably, regressing out the respiratory signal does not abolish the RSFC measured by rsfMRI data, suggesting that the respiration does not account for all the effects of the RSFC measured. *Figure 3—figure supplement 2* shows the differences of ACC seedmaps and ACC gamma activity-derived correlation maps before and after regressing out the RVT. Although RVT regression reduced ACC RSFC particularly in brain regions involved in the respiration network as expected, the major RSFC pattern remains consistent, indicating that a dominant component of RSFC is not ascribed by the respiration effect.

## Respiration and arousal changes

It is possible that the neural contribution to the respiration-related brain network is mediated by arousal changes, which is associated with both neural excitability and respiration (*Yackle et al., 2017*; *Shea, 1996*; *Fan et al., 2012*). The cerebral blood flow can be regulated by the innervation from the basal forebrain and locus coeruleus, and both regions are directly related to arousal changes (*Yackle et al., 2017*; *Lecrux and Hamel, 2016*; *Melnychuk et al., 2018*). Previous studies in humans demonstrated correlations among the fMRI signal, low-frequency fluctuations of respiration, and EEG alpha power (*Yuan et al., 2013*). The fMRI–respiration correlate was stronger at the eye-closing resting state with a greater arousal-level change than the eye-open state. In addition, strong association was identified between alpha wave and alertness level. These data together suggest that the correlations between low-frequency fluctuations of respiration, alpha EEG power, and the fMRI signal might be related to the fluctuation of wakefulness (*Yuan et al., 2013*). However, we do not believe the arousal level plays a dominant role in our results, given that our study was performed in anesthetized animals. Accordingly, we did not observe any coherence between alpha-band power and the slow variations of respiration, suggesting the respiration-related fMRI network we observed might be attributed to a different mechanism than arousal.

## The association between neural activity and respiration in human studies

The findings in the present study to a large extent echo the results reported in human studies. Previous research demonstrated bidirectional interplay between respiration and neural activity in humans. Neural activity in brainstem regions such as the preBötzinger complex directly controls the respiration (*Baertsch et al., 2021*; *Smith et al., 1991*). In addition, respiration-induced LFP oscillations were observed in multiple human brain regions such as the piriform cortex and hippocampus (*Zelano et al., 2016*; *Herrero et al., 2018*), consistent with our data. McKay et al. also found that voluntary hyperpnea was related to the neural activity of various brain regions including the primary sensory and motor cortices, supplementary motor area, caudate nucleus, and globus pallidum, which largely agree with the respiration-related brain network we identified in rats (*McKay et al., 2003*). Importantly, respiration-mediated neural activity was found to modulate brain function. For example, respiration is shown to modulate visual perceptual sensitivity, mediated by alpha power (*Kluger et al., 2021*). Respiration is also the key component in mindfulness meditation, which has been repeatedly shown to have modulatory effects on brain function (*Doll et al., 2016*). Furthermore, the coupling between neural activity and respiration may facilitate our understanding of brain disease. Hyam et al. stimulated the pedunculopontine region (PPNr) and observed improvement in upper airway function in Parkinson's patients (*Hyam et al., 2019*). Loss of neurons in the preBötzinger complex was found

to cause sleep-disordered breathing (*McKay et al., 2005*). Taken together, those findings indicate that the respiration-related brain network might be conserved across species and represent a general phenomenon in the mammalian brain, and this brain network might play an important role in normal and diseased brain function.

### Potential pitfalls

In the current study, we performed the experiment in lightly sedated animals to ensure the results were not confounded by the animal's motion. Indeed, animals' motion, particularly respiration-correlated motion, can induce systematic variation in the rsfMRI signal that potentially leads to an artifactual correlation pattern. Such a correlation pattern is difficult to be parceled out in our respiration-associated brain network. However, it has to be recognized that anesthesia might be a potential confounder by itself, as it may affect the respiration, as well as respiration-induced neural and vascular activities. For instance, anesthetics can reduce the respiration rate and affect LFP by increasing the low-frequency power and decreasing the high-frequency power (*Purdon et al., 2013*; *You et al., 2021*; *Bastos et al., 2021*). Our rationale for choosing combined low-dose isoflurane and dexmedetomidine as our anesthesia protocol is as follows: Isoflurane is a vasodilator, which can attenuate the BOLD signal (*Schwinn et al., 1990*), whereas the dexmedetomidine is a vasoconstrictor, which can counteract the vasodilatory effect from isoflurane (*Talke and Anderson, 2018*). This anesthesia protocol has been shown to minimize the confounding effects on the BOLD response and maintain the functional connectivity across both cortical and subcortical regions (*Grandjean et al., 2014*; *Benveniste et al., 2017*). Despite the potential confounding effects of anesthesia, previous studies demonstrated that similar respiration-coupled LFP oscillations were observed in both anesthetized and awake states (*Yanovsky et al., 2014*; *Lockmann et al., 2016*; *Nguyen Chi et al., 2016*). As the relationship between respiration and neural activity is preserved under anesthesia, our results obtained in anesthetized rats should remain valid. However, it has to be noted that the neural contributions in the respiration-related network may be different in signal amplitude and/or spatial pattern between the awake and anesthetized states, and other potential neural contributions, such as those mediated by emotional or arousal changes, may only be present in the awake state. Such factors warrant further investigation in awake animal studies. Another potential pitfall is that the number of animals in the present study is limited, and the reported effects are dominated by the number of scans/imaging sessions, rather than the number of individual rats. However, as main results are highly reproducible across animals as demonstrated in Results, we believe this issue should not be a major concern.

## Materials and methods

### Animals

The present study was approved by the Pennsylvania State University Institutional Animal Care and Use Committee (IACUC, protocol #: PRAMS201343583). Seven adult male Long–Evans rats (300–500 g) were used. Rats were housed in Plexiglas cages with food and water provided ad libitum. The ambient temperature was controlled at 22–24°C under a 12 hr light:12 hr dark schedule.

### Surgery

Stereotaxic surgeries were performed to implant electrodes in animals for the electrophysiology recording. The rat was anesthetized with an injection of ketamine (40 mg/kg) and xylazine (12 mg/kg), and remained anesthetized throughout the surgery by 0.75% isoflurane delivered through an endo-tracheal catheter intubated (PhysioSuite, Kent Scientific Corporaition). Antibiotics baytril (2.5 mg/kg) and long-acting analgesic drug buprenorphine were intramuscularly administered. During surgery, the temperature was monitored and maintained by a warming pad (PhysioSuite, Kent Scientific Corporaition). Heart rate and SpO$_2$ were monitored with a pulse oximetry (MouseSTAT Jr, Kent Scientific Corporation). An MR-compatible electrode (MRCM16LP, NeuroNexus Inc) was unilaterally implanted into the ACC (coordinates: anterior/posterior +1.5, medial/lateral +0.5, dorsal/ventral −2.8). The reference wire and grounding wire from the electrode were both connected to a silver wire placed in the cerebellum. After surgery, the animal was returned to the home cage and allowed to recover for at least 1 week.

## Simultaneous rsfMRI, respiration, and electrophysiology recordings

rsfMRI experiments were performed on a 7T Bruker 70/30 BioSpec running ParaVision 6.0.1 (Bruker, Billerica, MA) with a homemade single loop surface coil at the high-field MRI facility at the Pennsylvania State University. T2*-weighted gradient-echo rsfMRI images were acquired using an echo planar imaging sequence with following parameters: repetition time (TR) = 1 s; echo time (TE) = 15 ms; matrix size = 64 × 64; field of view = 3.2 × 3.2 cm$^2$; slice number = 20; slice thickness = 1 mm; volume number = 1200. T2-weighted structural images were also obtained using a rapid acquisition with relaxation enhancement (RARE) sequence with the following parameters: TR = 3000 ms; TE = 40 ms; matrix size = 256 × 256; field of view = 3.2 × 3.2 cm$^2$; slice number = 20; slice thickness = 1 mm; repetition number = 6.

During imaging, the animal was maintained in one of two anesthetized states: light sedation and isoelectric state. For imaging sessions under light sedation, animals were anesthetized with the combination of dexmedetomidine (initial bolus of 0.05 mg/kg followed by constant infusion at 0.1 mg kg$^{-1}$ hr$^{-1}$) and isoflurane (0.3%) (*Grandjean et al., 2014*), and spontaneous respiration was maintained in animals. For imaging sessions under the isoelectric state, sodium pentobarbital was administered with a 30 mg/kg bolus followed by continuous infusion (70 mg kg$^{-1}$ hr$^{-1}$) (*Du et al., 2008*). Before imaging, the rat was intubated via tracheal, and the respiration was controlled by a ventilator (PhysioSuite, Kent Scientific Corporaition) throughout the entire imaging session. Three rats were in the light sedation group with a total of 51 scans and four rats were in the isoelectric state group with a total of 48 scans. During all imaging sessions, the temperature was measured by a rectal thermometer and maintained at 37°C with warm air. Eyes of the animal were protected from dryness using artificial tear.

During rsfMRI acquisition, the respiration signal was simultaneously recorded at the sampling rate of 225 Hz by a respiration sensor placed under the animal's chest. Electrophysiology recording started 10 min before the beginning of rsfMRI acquisition and continued throughout the whole imaging session using a TDT recording system including an MR-compatible LP16CH headstage, PZ5 neurodigitizer amplifier, RZ2 BioAmp Processor and WS8 workstation (Tucker Davis Technologies Inc, Alachua, FL). The electrophysiology signal was sampled at 24,414 Hz and the unfiltered raw signal was used for further data processing.

## Data preprocessing

rsfMRI data were preprocessed using a pipeline described in our previous publications (*Liang et al., 2017*; *Tu et al., 2021b*). Briefly, rsfMRI images were first motion scrubbed based on relative framewise displacement (FD). Volumes with FD >0.25 mm and their adjacent preceding and following volumes were removed. Subsequently, data were preprocessed by performing motion correction (SPM12), co-registration to a defined atlas, spatial smoothing, as well as voxel-wise nuisance regression of motion parameters and the signals of the white matter and ventricles.

Electrophysiology data were preprocessed to remove MR artifacts using a template regression method (*Logothetis et al., 2001*; *Pan et al., 2011*). Specifically, raw electrophysiology signal was first temporally aligned with the corresponding rsfMRI scan. The potential phase differences across 16 electrophysiology channels were corrected by calculating cross-correlations of electrophysiology time series between channels, and the corrected signals from all 16 channels were summed. This summed signal was then segmented for each individual rsfMRI slice acquisition. Subsequently, an MRI interference template for each fMRI slice was estimated by averaging raw electrophysiology data across all segments corresponding the same slice acquisition from all rsfMRI volumes. The template was then aligned to each slice acquisition using cross-correlation. The final templates of all slices were linearly regressed out from the raw electrophysiology data to remove MR-induced artifacts, followed by a series of notch filters for harmonics of the power supply (60 Hz and multiples of 60 Hz) and slice acquisition (20 Hz and multiples of 20 Hz). Lastly, the continuous LFP was bandpass filtered (0.1–300 Hz). *Figure 1—figure supplement 2* shows an example of raw electrophysiology signals before denoising of MRI artifacts (*Figure 1—figure supplement 2A*), an example of the MRI artifact template (*Figure 1—figure supplement 2B*), as well as the LFP signal after MRI artifact denoising (*Figure 1—figure supplement 2C*).

The LFP spectrogram was computed using the MATLAB function *spectrogram* with window size = 1 s, step size = 0.1 s, as shown in *Figure 1E*.

## Data analysis

To determine the relationship between slow variations of respiration and the rsfMRI signal, Pearson correlation was voxel-wise calculated between the rsfMRI signal and RVT, which was calculated by the difference between consecutive peaks of inspiration and expiration divided by the time interval between the two peaks (*Birn et al., 2006*). One-sample *t*-test was performed to determine the statistical significance of correlations, thresholded at p < 0.05 after FDR correction of multiple comparisons.

To determine the artifactual impact of slow respiration variations on the rsfMRI signal, we convolved the RVT with the RRF, and then voxel-wise correlated it to the rsfMRI signal. The RRF is defined below:

$$\text{RFF}(t) = 0.6t^{2.1}e^{t/1.6} - 0.0023t^{3.54}e^{t/4.25}$$

The fast cyclic respiratory effects on the rsfMRI signal were quantified using the method RETRO-ICOR (*Glover et al., 2000*). Sinusoids were constructed based on TR relative to the phase of the respiratory cycles. For each brain voxel, the predicted fast respiration regressor was estimated by the linear combination of sinusoids that best fits to the voxel time series.

The phase-locking relationship between the respiratory rhythm and the LFP oscillations was determined by calculating the magnitude squared coherence between the two signals. We also assessed the relationship between the frequency band-specific power of the LFP signal and slow variations of respiration. LFP band power was obtained using the MATLAB function *spectrogram* with window size = 1 s, step size = 0.1 s based on the conventional LFP band definition from previous studies (delta: 1–4 Hz, theta: 4–7 Hz, alpha: 7–13 Hz, beta: 13–30 Hz, gamma: 40–100 Hz) (*Lu et al., 2016*; *Magri et al., 2012*). The magnitude squared coherence and the phase relationship between the power of each LFP band and RVT were calculated using the MATLAB function *mscohere*.

To determine the relationship between the gamma-band power and rsfMRI signal, the time course of gamma power was first convolved with the hemodynamic response function (HRF, a single gamma probability distribution function [*a* = 3, *b* = 0.8]; *Liang et al., 2017*), and voxel-wise Pearson correlations between the HRF-convolved gamma power and rsfMRI signals in the brain were computed.

The ACC seedmap was obtained by voxel-wise calculating the Pearson correlations between the regionally averaged rsfMRI time course of the unilateral right-side ACC with the rsfMRI signals across the whole brain. To perform the region of interest (ROI)-based analysis, the whole brain was parcellated into 132 anatomical ROIs based on the *Swanson, 2004* Atlas. The whole-brain RSFC matrix was computed by Pearson correlations of regionally averaged rsfMRI time courses between pairwise ROIs.

## Acknowledgements

We thank Dr. Yuncong Ma and Ms. Xiaoai Chen for their technical support, and Dr. Xiao Liu for scientific discussion. The present study was partially supported by National Institute of Neurological Disorders and Stroke (R01NS085200), National Institute of Mental Health (RF1MH114224), and National Institute of General Medical Sciences (R01GM141792). The content is solely the responsibility of the authors and does not necessarily represent the official views of the National Institutes of Health.

## Additional information

### Funding

| Funder | Grant reference number | Author |
|---|---|---|
| National Institute of Neurological Disorders and Stroke | R01NS085200 | Nanyin Zhang |
| National Institute of Mental Health | RF1MH114224 | Nanyin Zhang |
| National Institute of General Medical Sciences | R01GM141792 | Nanyin Zhang |

| Funder | Grant reference number | Author |
|---|---|---|

The funders had no role in study design, data collection, and interpretation, or the decision to submit the work for publication.

## Author contributions
Wenyu Tu, Data curation, Formal analysis, Validation, Investigation, Methodology, Writing – original draft; Nanyin Zhang, Conceptualization, Supervision, Funding acquisition, Validation, Investigation, Writing – original draft, Project administration, Writing – review and editing

## Author ORCIDs
Wenyu Tu http://orcid.org/0000-0003-3480-2098
Nanyin Zhang http://orcid.org/0000-0002-5824-9058

## Ethics
The present study was approved by the Pennsylvania State University Institutional Animal Care and Use Committee (IACUC) with the protocol number of PRAMS201343583.

## Decision letter and Author response
Decision letter https://doi.org/10.7554/eLife.81555.sa1
Author response https://doi.org/10.7554/eLife.81555.sa2

# Additional files

## Supplementary files
• MDAR checklist

## Data availability
All data for this study have been deposited to NITRIC repository.

The following dataset was generated:

| Author(s) | Year | Dataset title | Dataset URL | Database and Identifier |
|---|---|---|---|---|
| Tu W, Zhang N | 2022 | Electrophysiology, resting state fMRI and respiration in rats | https://www.nitrc.org/docman/?group_id=1582 | NeuroImaging Tools and Resources Collaboratory (NITRC), 1582 |

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
