## [Editor Report]

This paper will be of interest to researchers studying control of respiration and also those developing functional magnetic resonance imaging methodology. The work provides insight into the relationship between brain activity (measured directly) and non-invasive functional magnetic resonance imaging measures. The authors find that the respiration signal is associated with the γ band in the cingulate cortex, and both the γ signal and respiration signal correlate with distributed neuronal networks across the brain. This contributes to our knowledge of the contribution of respiration on neuro and neuro-vascular signals during resting conditions.

---

## [Decision Letter]

**Decision letter after peer review:**

Thank you for submitting your article "Respiration-driven brain network: neural underpinning of breathing correlates with resting-state fMRI signal" for consideration by *eLife*. Your article has been reviewed by 3 peer reviewers, and the evaluation has been overseen by a Reviewing Editor and Timothy Behrens as the Senior Editor. The following individual involved in review of your submission has agreed to reveal their identity: Joanes Grandjean (Reviewer #1).

Essential revisions:

1) Tempering of language purporting to establish a causal relationship between respiration and the observed γ oscillations.

2) Comment on the relatively small number of rodents used in the study. [In consultation between the reviewers, it has been pointed out that the reported effects are based on statistical tests driven by the number of timepoints, rather than the number of individual rodents.]

3) Discussion of the potential effects of anaesthesia on the study conclusions.

4) Placing results in the context of related human studies.

5) Analysis of RVT estimates (variance) in the isoelectric state.

*Reviewer #1 (Recommendations for the authors):*

As indicated in the public review, I don't find anything that is lacking in the study. I find it compelling, and I do not wish to burden the authors with additional work that may only remotely nuance their excellent work.

If anything, I am not sure the statement below (and other similar later in the discussion) are sufficiently substantiated. Γ oscillations are correlated with respiration in this study. Other tests to further establish causality would be required, such as comparing spontaneous vs mechanical respiration. Personally, I can think of other mechanisms to explain the results, such as predictive encoding of respiration-associated outputs in the cingulate area.

"These results suggest that respiration can drive γ oscillations, which further lead to BOLD signal changes in the respiration network"

*Reviewer #2 (Recommendations for the authors):*

1) The sample size is small -- the study hasn't been pre-registered nor has a formal sample size calculation been provided. This does mean that a number of analyses could have been tried before coming out with this result. With calls for ever increasing numbers of participants in human studies, a sample size of 7 seems difficult to justify.

2) Stronger justification for the use of the ACC as electrode location is essential. The Liotti paper gave CO2 inhalation to healthy humans and reported which areas of the brain showed increased cerebral blood flow, but interpreted the findings in the context of breathlessness. There is a lot more recent work from a range of laboratories done in a much more controlled manner that would justify better the use of the ACC as a component of the respiratory network. There are other areas consistently identified in respiration (e.g. anterior insula, prefrontal cortex etc) that could have been chosen so it's important here not to make the conclusions too ACC specific (in the most part the authors have done this)

3) The effects of anaesthesia are, in my view, a major potential confound. In addition to direct effects on BOLD responsiveness, isoflurane, pentobarbital and (particularly) buprenorphine will depress respiration, raising arterial CO2 and dampening BOLD responsiveness. Ketamine may stimulate respiration doing the opposite. Interaction effects between the different drugs are highly likely and difficult to predict how they would affect BOLD responses.

4) Why was RVT used as the measure of minute ventilation when it would have been possible to take direct measures of breathing via the endotracheal tube? This would be considerably more accurate.

5) Novelty – with such a small sample size it's difficult to know what is truly new. For example Alexander Green's group in Oxford report EEG changes (in α band) in patients with deep brain stimulator electrodes during various respiratory manoeuvres. Of course those studies are also limited by small sample size and diseased patients so comparison is difficult.

*Reviewer #3 (Recommendations for the authors):*

1) As mentioned in the public review, the statistics of RVT in the isoelectric condition are not reported (unless I have missed it), so it is not clear whether the corresponding decrease in correlation with fMRI could be due to reduced temporal variation in RVT. I would therefore recommend reporting the magnitude of respiratory volume and rate variation and the RVT power spectrum, comparing both the light anesthesia and isoelectric states, in order to determine if this is the case. If RVT is indeed found to be less variable in this condition, it would strengthen the study if the authors are able to perform additional experiments that induce variations in breathing (if that is a possibility).

2) Relatedly, further support could be provided if the authors observe that the isoelectric state selectively disrupts the potential neuronal component (instantaneous fMRI correlation with RVT), but preserves the non-neural CO2 effect on BOLD (assessed by convolving RVT with the respiration response function).

3) To my understanding, RVT was correlated with fMRI at zero time-lag. However, when γ power was correlated with fMRI, it was first convolved with a standard hemodynamic response function. It was not clear to me why RVT was also not convolved with a hemodynamic response function to assess potential neural correlations with fMRI, especially given that it has highest coherence with the γ power signal at zero phase lag.

4) It is shown that the fMRI spatial maps that correlate with ACC γ power resemble the maps of the seed-based correlation from the ACC fMRI signal. For completeness, it would be useful to also report the temporal correlation between the fMRI and γ power signals measured in the ACC.

5) It is mentioned on p.2, line 85, that anesthesia was used to ensure that results are not confounded by motion. Though animals were only lightly sedated, it would be helpful to explain in the discussion whether this level of anesthesia might also impact fMRI and/or electrophysiological signals, to help readers better interpret the findings.

6) One analysis supporting the possible neural origin of RVT correlation in fMRI data is that regressing out neural (γ-band LFP) signals reduced the correlation between RVT and fMRI. This is indeed promising, though I suppose that if there are any non-neural hemodynamic effects of respiration on brain vasculature closer to zero lag (and which therefore also correlate with γ LFP), these would also be reduced when regressing out γ LFP. It would be helpful to include a discussion of whether any non-neural hemodynamic effects in fMRI could occur at zero lag.

[Editors' note: further revisions were suggested prior to acceptance, as described below.]

Thank you for resubmitting your work entitled "Neural underpinning of a respiration-associated resting-state fMRI network" for further consideration by *eLife*. Your revised article has been evaluated by Timothy Behrens (Senior Editor) and a Reviewing Editor.

The manuscript has been improved but there are some remaining issues that need to be addressed, as outlined below:

In the Response to Reviewers, the authors present arguments that their anaesthesia protocol has minimal disruption on neurovascular coupling. Unfortunately, though, this does not come through in the discussion added to the paper itself, which focuses on respiration-LFP coupling. Given that the effects of anaesthesia on neurovascular coupling were very specifically raised by Reviewers 2 and 3 (and will be shared by many readers), please add a brief discussion along the lines of the response to Reviewer 2 Comment 3.

---

## [Author Response]

Essential revisions:1) Tempering of language purporting to establish a causal relationship between respiration and the observed γ oscillations.

We appreciate and agree with this suggestion. In the revised manuscript, we modified all related statements, including the title, from establishing a causal relationship to an associative relationship between respiration and neural activity.

2) Comment on the relatively small number of rodents used in the study. [In consultation between the reviewers, it has been pointed out that the reported effects are based on statistical tests driven by the number of timepoints, rather than the number of individual rodents.]

The reviewer’s comment is well accepted. We completely acknowledge that the number of animals in the present study is limited, and the reported effects are dominated by the number of scans/imaging sessions, rather than the number of individual rats. In our revised manuscript, we added statements to discuss this issue.

We would like to further clarify that the number of animals used was based on the conventional sample size of electrophysiology-fMRI experiments, typically in the range of 5-10 animals ^1-5^ (7 animals in our study). In electrophysiology recordings, a large number of data points are collected and the results are generally reproducible across animals. To compensate for lower sensitivity/higher variability of fMRI data, unlike conventional fMRI studies, here in each animal we acquired a great number of scans on separate days (total scan number = 99, ~14 scans per animal, 1200 fMRI volumes per scan, 51 scans at the light sedation state and 48 scans at the isoelectric state). This method can significantly reduce the variability of results obtained from individual animals. Below are four figures we added to the revised manuscript demonstrating major results from individual animals. Indeed, all major findings are reproducible across animals, and therefore, we believe this method can significantly mitigate the issue of the small sample size of animals in our study.

3) Discussion of the potential effects of anaesthesia on the study conclusions.

Thanks for the comment. In the revised manuscript, we added a paragraph to discuss the potential effects of anesthesia on our results, quoted below.

“In the current study, we performed the experiment in lightly sedated animals to ensure the results were not confounded by the animal’s motion. Animals’ motion, particularly respiration-correlated motion, can induce systematic variation in the rsfMRI signal that potentially leads to an artefactual correlation pattern. Such a correlation pattern is difficult to be parceled out in our respiration-associated brain network. However, it has to be recognized that anesthesia might be a potential confounder by itself, as it may affect the respiration, as well as respiration-induced neural and vascular activities. For instance, anesthetics can reduce the respiration rate and affect LFP by increasing the low frequency power and decreasing the high frequency power ^6-8^. However, previous studies have demonstrated that similar respiration-coupled LFP oscillations were observed in both anesthetized and awake states ^9-11^. As the relationship between respiration and neural activity is preserved under anesthesia, our results obtained in anesthetized rats should remain valid. It has to be noted that the neural contributions in the respiration-related network may be different in signal amplitude and/or spatial pattern between the awake and anesthetized states, and other potential neural contributions, such as those mediated by emotional or arousal changes, may only be present in the awake state. Such factors warrant further investigation in awake animal studies.”

4) Placing results in the context of related human studies.

We added more discussions of our results in the context of related human studies, quoted below.

“The findings in the present study to a large extent echo the results reported in human studies. Previous research demonstrated bidirectional interplay between respiration and neural activity in humans. Neural activity in brainstem regions such as the preBötzinger complex directly controls the respiration^12,13^. In addition, respiration-induced LFP oscillations were observed in multiple human brain regions such as the piriform cortex and hippocampus^14,15^, consistent with our data. McKay and colleagues also found that voluntary hyperpnea was related to the neural activity of various brain regions including the primary sensory and motor cortices, supplementary motor area, caudate nucleus, and globus pallidum, which largely agree with the respiration-related brain network we identified in rats^16^. Importantly, respiration-mediated neural activity was found to modulate brain function. For example, respiration is shown to modulate visual perceptual sensitivity, mediated by α power^17^. Respiration is also the key component in mindfulness meditation, which has been repeatedly shown to have modulatory effects on brain function^18^. Furthermore, the coupling between neural activity and respiration may facilitate our understanding of brain disease. Hyam *et al.* stimulated the pedunculopontine region (PPNr) and observed improvement in upper airway function in Parkinson’s patients^19^. Loss of neurons in the preBötzinger complex was found to cause sleep-disordered breathing^20^. Taken together, those findings indicate that the respiration-related brain network might be conserved across species and represent a general phenomenon in the mammalian brain, and this brain network might play an important role in normal and diseased brain function.”

5) Analysis of RVT estimates (variance) in the isoelectric state.

We greatly appreciate the suggestion. We calculated the respiration rate (Figure 5D) as well as the standard deviation (SD) and power spectrum of RVT across animals. We did not observe any significant difference in the SD of the respiration rate (Figure 5E, two-sample t-test, t = -1.0, p = 0.32) or the SD of RVT (Figure 5F, two-sample t-test, t = -0.87, p = 0.39) between the light-sedation and isoelectric states, suggesting that the temporal variance of respiration is similar between the two states. Interestingly, we observed a characteristic 1/f pattern in the RVT power during light sedation, but this pattern was not present at the isoelectric state (Figure 5G). This difference is consistent to the ACC BOLD signal power spectra in the light-sedation and isoelectric states (Figure 5I). The new results and figure are added to the revised manuscript.

Reviewer #1 (Recommendations for the authors):As indicated in the public review, I don't find anything that is lacking in the study. I find it compelling, and I do not wish to burden the authors with additional work that may only remotely nuance their excellent work.If anything, I am not sure the statement below (and other similar later in the discussion) are sufficiently substantiated. Γ oscillations are correlated with respiration in this study. Other tests to further establish causality would be required, such as comparing spontaneous vs mechanical respiration. Personally, I can think of other mechanisms to explain the results, such as predictive encoding of respiration-associated outputs in the cingulate area."These results suggest that respiration can drive γ oscillations, which further lead to BOLD signal changes in the respiration network"

We really appreciate the encouraging comment from the reviewer. The reviewer’s suggestion is well accepted. In the revised manuscript, we modified all the related statements, including the title, from establishing a causal relationship to an associative relationship between respiration and neural activity. Below is the revised statement for the one the reviewer quoted. Other revised statements are highlighted in red in the revised manuscript.

“These results suggest that respiration is associated with γ oscillations, which is necessary for BOLD signal changes in the respiration network.”

Reviewer #2 (Recommendations for the authors):1) The sample size is small -- the study hasn't been pre-registered nor has a formal sample size calculation been provided. This does mean that a number of analyses could have been tried before coming out with this result. With calls for ever increasing numbers of participants in human studies, a sample size of 7 seems difficult to justify.

Please refer to the response to Comment 2 in the section of Essential revisions from Reviewing Editor.

2) Stronger justification for the use of the ACC as electrode location is essential. The Liotti paper gave CO2 inhalation to healthy humans and reported which areas of the brain showed increased cerebral blood flow, but interpreted the findings in the context of breathlessness. There is a lot more recent work from a range of laboratories done in a much more controlled manner that would justify better the use of the ACC as a component of the respiratory network. There are other areas consistently identified in respiration (e.g. anterior insula, prefrontal cortex etc) that could have been chosen so it's important here not to make the conclusions too ACC specific (in the most part the authors have done this)

We appreciate the reviewer’s suggestion. In the revised manuscript, we added more rationale and references from recent studies supporting the vital role of the ACC in the respiration network, quoted below.

“The ACC was selected given its critical role in respiratory modulation, evidenced by early studies showing that ACC was activated during breathlessness^21,22^. More recent research further demonstrates the role of ACC in respiratory control^23-25^. For example, electrophysiology recordings in humans showed that ACC exhibited different neural responses to separate respirational tasks including breath-holding, voluntary deep breathing and hypercapnia^25^. In addition, the ACC is a key region in the rodent default-mode network (DMN) ^26,27^, which has been linked to respiration-related fMRI signal changes ^28^.”

We completely agree with the reviewer that a number of other regions are also involved in the respiration network, and our results are indeed not ACC specific. In fact, the main finding of our paper is that we identified a network of brain regions whose neural activities are associated with respiration (also evident from the title), and ACC is one node of this network, through which the network is identified.

3) The effects of anaesthesia are, in my view, a major potential confound. In addition to direct effects on BOLD responsiveness, isoflurane, pentobarbital and (particularly) buprenorphine will depress respiration, raising arterial CO2 and dampening BOLD responsiveness. Ketamine may stimulate respiration doing the opposite. Interaction effects between the different drugs are highly likely and difficult to predict how they would affect BOLD responses.

We appreciate this valuable comment. Please refer to the response to Comment 3 in the section of Essential revisions from Reviewing Editor for our general response to the potential impact of anesthesia. Below are the more specific responses to the potential effects of different anesthetic drugs the reviewer mentioned.

First, we would like to clarify that we only used the combination of low-dose isoflurane and low-dose dexmedetomidine to induce the light sedation state in animals during simultaneous fMRI, electrophysiology and respiration recordings. Our rationale for choosing this anesthesia is as follows: isoflurane is a vasodilator, which can attenuate the BOLD signal ^29^, whereas the dexmedetomidine is a vasoconstrictor, which can counteract the vasodilatory effect from isoflurane^30^. This type of anesthesia has been shown to minimize the confounding effects in the BOLD response and maintain the functional connectivity across both cortical and subcortical regions^31,32^. Pentobarbital was used in separate experiments to induce the isoelectric state. This control experiment allows us to determine the necessary role of neural activity in the respiration-related brain network identified in our study.

Ketamine and buprenorphine were not used in any experiment, but only used during the surgery for electrode implantation. The animals were allowed to recover for at least one week after surgery. The drugs should have been fully metabolized by the time of fMRI imaging, so effects from those drugs would not affect our results.

Overall, we agree with the reviewer that anesthesia is a potential confound. In the revised manuscript we have discussed this potential pitfall, quoted below.

“In the current study, we performed the experiment in lightly sedated animals to ensure the results were not confounded by the animal’s motion. Indeed, animals’ motion, particularly respiration-correlated motion, can induce systematic variation in the rsfMRI signal that potentially leads to an artefactual correlation pattern. Such a correlation pattern is difficult to be parceled out in our respiration-associated brain network. However, it has to be recognized that anesthesia might be a potential confounder by itself, as it may affect the respiration, as well as respiration-induced neural and vascular activities. For instance, anesthetics can reduce the respiration rate and affect local field potential by increasing the low-frequency power and decreasing the high-frequency power^6-8^. However, previous studies demonstrated that similar respiration-coupled LFP oscillations were observed in both anesthetized and awake states^9-11^. As the relationship between respiration and neural activity is preserved under anesthesia, our results obtained in anesthetized rats should remain valid. Notably, the neural contributions in the respiration-related network may be different in signal amplitude and/or spatial pattern between the awake and anesthetized states, and other potential neural contributions, mediated by emotional or arousal changes, are only present in awake state. Such factors warrant further investigation in future awake animal studies.”

4) Why was RVT used as the measure of minute ventilation when it would have been possible to take direct measures of breathing via the endotracheal tube? This would be considerably more accurate.

We agree with the reviewer that respiration can also be measured via the endotracheal tube in animals that were ventilated in the isoelectric state. However, the respiration in animals during light sedation, which were not intubated, can only be measured using the respiration pads. As we need to maintain a consistent way to measure respiration across conditions for comparison, we used the respiration pad in our study. This method also provides an accurate measurement of respiration and RVT ^33,34^.

5) Novelty – with such a small sample size it's difficult to know what is truly new. For example Alexander Green's group in Oxford report EEG changes (in α band) in patients with deep brain stimulator electrodes during various respiratory manoeuvres. Of course those studies are also limited by small sample size and diseased patients so comparison is difficult.

The major novelty of the present study is the finding of a network of brain regions whose neural activities are associated with respiration. We achieved this goal by simultaneously recording electrophysiology, whole-brain fMRI, and respiration signals. This is distinct from most other human (or animal) studies that only focused on the role of a (or a few) specific brain region(s). For instance, the study by Alexander Green's group mentioned by reviewer ^19^ found that stimulation of the pedunculopontine region (PPNr) improved upper airway function in parkinson’s disease patients. The α power in PPNr increased during forced respiratory maneuvers. Their study is important for understanding the potential role of neural activity of one region (PPNr) in respiration control, which is very different from our study.

In the revised manuscript, we added the discussion for our results in relation to human studies, quoted below:

“The findings in the present study to a large extent echo the results reported in human studies. Previous research demonstrated bidirectional interplay between respiration and neural activity in humans. Neural activity in brainstem regions such as the preBötzinger complex directly controls the respiration^12,13^. In addition, respiration-induced LFP oscillation was observed in multiple human brain regions such as the piriform cortex and hippocampus^14,15^, which is consistent with our data. McKay and colleagues also found that voluntary hyperpnea was related to the neural activity of various brain regions including the primary sensory and motor cortices, supplementary motor area, caudate nucleus, and globus pallidum, which largely agreeing with the respiration-related brain network we identified in rats^16^. Importantly, respiration-mediated neural activity was found to modulate brain function. For example, respiration is shown to modulate visual perceptual sensitivity, mediated by α power^17^. Respiration is also the key component in mindfulness meditation, which has been repeatedly shown to have modulatory effects on brain function ^18^. Furthermore, the coupling between neural activity and respiration may facilitate our understanding of brain disease. Hyam *et al.* stimulated the pedunculopontine region (PPNr) and observed improvement in upper airway function in Parkinson’s patients ^19^. Loss of neurons in the preBötzinger complex was found to cause sleep-disordered breathing ^20^. Taken together, those findings indicate that the respiration-related brain network might be conserved across species and represent a general phenomenon in the mammalian brain, and this brain network might play an important role in normal and diseased brain function.”

Reviewer #3 (Recommendations for the authors):1) As mentioned in the public review, the statistics of RVT in the isoelectric condition are not reported (unless I have missed it), so it is not clear whether the corresponding decrease in correlation with fMRI could be due to reduced temporal variation in RVT. I would therefore recommend reporting the magnitude of respiratory volume and rate variation and the RVT power spectrum, comparing both the light anesthesia and isoelectric states, in order to determine if this is the case. If RVT is indeed found to be less variable in this condition, it would strengthen the study if the authors are able to perform additional experiments that induce variations in breathing (if that is a possibility).

Please refer to the response to Comment 5 in the section of Essential Revisions from Reviewing Editor.

2) Relatedly, further support could be provided if the authors observe that the isoelectric state selectively disrupts the potential neuronal component (instantaneous fMRI correlation with RVT), but preserves the non-neural CO2 effect on BOLD (assessed by convolving RVT with the respiration response function).

This is a very interesting point. The non-neural CO2 effect on the BOLD signal in the isoelectric state, estimated by convolving RVT with the respiration response function and then voxel-wise correlating to the BOLD signal, appeared to be weaker than the light-sedation state (Figure 4 —figure supplement 3), as shown below (Author response image 1, left). This is likely attributed to the fact that the metabolic rate was much lower in the isoelectric state (approximately half) ^35^, resulting in weaker fluctuations in the CO2 level during respiration, which can explain less appreciable CO2 effect on the BOLD signal in the isoelectric state. However, statistical comparison of the respiration artifactual pattern between the light-sedation and isoelectric states did not reveal any significant difference (Author response image 1, middle, t-test, threshold p = 0.05, FDR corrected), in sharp contrast to the statistical comparison of the respiration network pattern between the light-sedation and isoelectric states (Author response image 1, right, t-test, p < 0.05, FDR corrected). These data indicate that the artefactual effects of respiration were more or less similar between the two states, but the respiration-related brain network activity was substantially suppressed at the isoelectric state.

**Author response image 1. sa2fig1:** Left, voxel-wise correlations between the estimated time course of respirational artifact, estimated by convolving RVT with the respiration response function, and rsfMRI signals at the isoelectric state. The same artifactual pattern during light sedation is shown in Figure 4 —figure supplement 3. Middle, statistical comparison of the respiration artifactual patterns between light sedation and isoelectric states (t-test, threshold p = 0.05, FDR corrected). Right, statistical comparison of the respiration-related brain network patterns between light sedation and isoelectric states (t-test, p < 0.05, FDR corrected).

3) To my understanding, RVT was correlated with fMRI at zero time-lag. However, when γ power was correlated with fMRI, it was first convolved with a standard hemodynamic response function. It was not clear to me why RVT was also not convolved with a hemodynamic response function to assess potential neural correlations with fMRI, especially given that it has highest coherence with the γ power signal at zero phase lag.

The hemodynamic response function (HRF) referred herein by definition represents the transfer function between neural activity and fMRI signal. It is effectively a low-pass filter that transfers fast neural activity to slower fMRI signal. Because HRF is not the transfer function between RVT and fMRI signal, it would not be appropriate to convolve RVT with HRF. In fact, as the RVT itself represents low-frequency fluctuations of respiration (the peak coherence between RVT and γ power is at a low frequency (0.035Hz)), convolving RVT with HRF should not significantly alter the RVT waveform. Indeed, in Author response image 2 we show the voxel-wise correlation pattern between rsfMRI signal and the time course after convolving RVT with HRF. The pattern is very similar to the RVT-rsfMRI correlation pattern we obtained (Figure 4B).

**Author response image 2. sa2fig2:** Voxel-wise correlations between rsfMRI signals and the time course obtained by convolving RVT with HRF.

4) It is shown that the fMRI spatial maps that correlate with ACC γ power resemble the maps of the seed-based correlation from the ACC fMRI signal. For completeness, it would be useful to also report the temporal correlation between the fMRI and γ power signals measured in the ACC.

Thanks for your suggestion, we have reported the correlation between fMRI signal of ACC and γ power (r = 0.086, p = 0.003) in the revised manuscript.

5) It is mentioned on p.2, line 85, that anesthesia was used to ensure that results are not confounded by motion. Though animals were only lightly sedated, it would be helpful to explain in the discussion whether this level of anesthesia might also impact fMRI and/or electrophysiological signals, to help readers better interpret the findings.

Please refer to our response to Comment 3 in the section of Essential Revisions from Reviewing Editor, as well as our response to Comment 3 for Reviewer #2 for the issue of the potential impact of anesthesia.

6) One analysis supporting the possible neural origin of RVT correlation in fMRI data is that regressing out neural (γ-band LFP) signals reduced the correlation between RVT and fMRI. This is indeed promising, though I suppose that if there are any non-neural hemodynamic effects of respiration on brain vasculature closer to zero lag (and which therefore also correlate with γ LFP), these would also be reduced when regressing out γ LFP. It would be helpful to include a discussion of whether any non-neural hemodynamic effects in fMRI could occur at zero lag.

This is an interesting point. Although we cannot think of any non-neural hemodynamic effects of respiration on the rsfMRI signal that could occur at zero lag, we do agree with the reviewer that regressing out γ LFP will reduce these effects. We have discussed this possibility in our revised manuscript, quoted below.

“This notion is further supported by our data that regressing out γ-band LFP signals tremendously reduced the correlation between RVT and fMRI (Figure 4D-E), although it is still possible that if there are non-neural hemodynamic effects of respiration on the rsfMRI signal that occur at zero lag (i.e. a non-neural hemodynamic factor that are synchronized with γ LFP), regressing out γ LFP could also reduce these effects.”

References

1. Angenstein, F. The role of ongoing neuronal activity for baseline and stimulus-induced BOLD signals in the rat hippocampus. *Neuroimage* 202, 116082, doi:10.1016/j.neuroimage.2019.116082 (2019).

2. Baek, K. *et al.* Layer-specific interhemispheric functional connectivity in the somatosensory cortex of rats: resting state electrophysiology and fMRI studies. *Brain Struct Funct* 221, 2801-2815, doi:10.1007/s00429-015-1073-0 (2016).

3. Bovet-Carmona, M. *et al.* Disentangling the role of TRPM4 in hippocampus-dependent plasticity and learning: an electrophysiological, behavioral and FMRI approach. *Brain Struct Funct* 223, 3557-3576, doi:10.1007/s00429-018-1706-1 (2018).

4. Jaime, S., Cavazos, J. E., Yang, Y. and Lu, H. Longitudinal observations using simultaneous fMRI, multiple channel electrophysiology recording, and chemical microiontophoresis in the rat brain. *J Neurosci Methods* 306, 68-76, doi:10.1016/j.jneumeth.2018.05.010 (2018).

5. Thompson, G. J., Pan, W. J. and Keilholz, S. D. Different dynamic resting state fMRI patterns are linked to different frequencies of neural activity. *J Neurophysiol* 114, 114-124, doi:10.1152/jn.00235.2015 (2015).

6. Purdon, P. L. *et al.* Electroencephalogram signatures of loss and recovery of consciousness from propofol. *Proc Natl Acad Sci U S A* 110, E1142-1151, doi:10.1073/pnas.1221180110 (2013).

7. You, T., Im, G. H. and Kim, S. G. Characterization of brain-wide somatosensory BOLD fMRI in mice under dexmedetomidine/isoflurane and ketamine/xylazine. *Sci Rep* 11, 13110, doi:10.1038/s41598-021-92582-5 (2021).

8. Bastos, A. M. *et al.* Neural effects of propofol-induced unconsciousness and its reversal using thalamic stimulation. *ELife* 10, doi:10.7554/*eLife*.60824 (2021).

9. Lockmann, A. L., Laplagne, D. A., Leao, R. N. and Tort, A. B. A Respiration-Coupled Rhythm in the Rat Hippocampus Independent of Theta and Slow Oscillations. *J Neurosci* 36, 5338-5352, doi:10.1523/JNEUROSCI.3452-15.2016 (2016).

10. Nguyen Chi, V. *et al.* Hippocampal Respiration-Driven Rhythm Distinct from Theta Oscillations in Awake Mice. *J Neurosci* 36, 162-177, doi:10.1523/JNEUROSCI.2848-15.2016 (2016).

11. Yanovsky, Y., Ciatipis, M., Draguhn, A., Tort, A. B. and Brankack, J. Slow oscillations in the mouse hippocampus entrained by nasal respiration. *J Neurosci* 34, 5949-5964, doi:10.1523/JNEUROSCI.5287-13.2014 (2014).

12. Baertsch, N. A., Bush, N. E., Burgraff, N. J. and Ramirez, J. M. Dual mechanisms of opioid-induced respiratory depression in the inspiratory rhythm-generating network. *ELife* 10, doi:10.7554/*eLife*.67523 (2021).

13. Smith, J. C., Ellenberger, H. H., Ballanyi, K., Richter, D. W. and Feldman, J. L. Pre-Botzinger complex: a brainstem region that may generate respiratory rhythm in mammals. *Science* 254, 726-729, doi:10.1126/science.1683005 (1991).

14. Herrero, J. L., Khuvis, S., Yeagle, E., Cerf, M. and Mehta, A. D. Breathing above the brain stem: volitional control and attentional modulation in humans. *J Neurophysiol* 119, 145-159, doi:10.1152/jn.00551.2017 (2018).

15. Zelano, C. *et al.* Nasal Respiration Entrains Human Limbic Oscillations and Modulates Cognitive Function. *J Neurosci* 36, 12448-12467, doi:10.1523/JNEUROSCI.2586-16.2016 (2016).

16. McKay, L. C., Evans, K. C., Frackowiak, R. S. and Corfield, D. R. Neural correlates of voluntary breathing in humans. *J Appl Physiol (1985)* 95, 1170-1178, doi:10.1152/japplphysiol.00641.2002 (2003).

17. Kluger, D. S., Balestrieri, E., Busch, N. A. and Gross, J. Respiration aligns perception with neural excitability. *ELife* 10, doi:10.7554/*eLife*.70907 (2021).

18. Doll, A. *et al.* Mindful attention to breath regulates emotions via increased amygdala-prefrontal cortex connectivity. *Neuroimage* 134, 305-313, doi:10.1016/j.neuroimage.2016.03.041 (2016).

19. Hyam, J. A. *et al.* The pedunculopontine region and breathing in Parkinson's disease. *Ann Clin Transl Neurol* 6, 837-847, doi:10.1002/acn3.752 (2019).

20. McKay, L. C., Janczewski, W. A. and Feldman, J. L. Sleep-disordered breathing after targeted ablation of preBotzinger complex neurons. *Nat Neurosci* 8, 1142-1144, doi:10.1038/nn1517 (2005).

21. Liotti, M. *et al.* Brain responses associated with consciousness of breathlessness (air hunger). *Proc Natl Acad Sci U S A* 98, 2035-2040, doi:10.1073/pnas.98.4.2035 (2001).

22. Evans, K. C. *et al.* BOLD fMRI identifies limbic, paralimbic, and cerebellar activation during air hunger. *J Neurophysiol* 88, 1500-1511, doi:10.1152/jn.2002.88.3.1500 (2002).

23. Evans, K. C. *et al.* Modulation of spontaneous breathing via limbic/paralimbic-bulbar circuitry: an event-related fMRI study. *Neuroimage* 47, 961-971, doi:10.1016/j.neuroimage.2009.05.025 (2009).

24. Tort, A. B. L. *et al.* Parallel detection of theta and respiration-coupled oscillations throughout the mouse brain. *Sci Rep* 8, 6432, doi:10.1038/s41598-018-24629-z (2018).

25. Holton, P. *et al.* Differential responses to breath-holding, voluntary deep breathing and hypercapnia in left and right dorsal anterior cingulate. *Exp Physiol* 106, 726-735, doi:10.1113/EP088961 (2021).

26. Lu, H. *et al.* Rat brains also have a default mode network. *Proc Natl Acad Sci U S A* 109, 3979-3984, doi:1200506109 [pii]10.1073/pnas.1200506109 (2012).

27. Tu, W., Ma, Z., Ma, Y., Dopfel, D. and Zhang, N. Suppressing Anterior Cingulate Cortex Modulates Default Mode Network and Behavior in Awake Rats. *Cereb Cortex* 31, 312-323, doi:10.1093/cercor/bhaa227 (2021).

28. Birn, R. M., Diamond, J. B., Smith, M. A. and Bandettini, P. A. Separating respiratory-variation-related fluctuations from neuronal-activity-related fluctuations in fMRI. *Neuroimage* 31, 1536-1548, doi:10.1016/j.neuroimage.2006.02.048 (2006).

29. Schwinn, D. A., McIntyre, R. W. and Reves, J. G. Isoflurane-induced vasodilation: role of the α-adrenergic nervous system. *Anesth Analg* 71, 451-459, doi:10.1213/00000539-199011000-00001 (1990).

30. Talke, P. and Anderson, B. J. Pharmacokinetics and pharmacodynamics of dexmedetomidine-induced vasoconstriction in healthy volunteers. *Br J Clin Pharmacol* 84, 1364-1372, doi:10.1111/bcp.13571 (2018).

31. Grandjean, J., Schroeter, A., Batata, I. and Rudin, M. Optimization of anesthesia protocol for resting-state fMRI in mice based on differential effects of anesthetics on functional connectivity patterns. *Neuroimage* 102 Pt 2, 838-847, doi:10.1016/j.neuroimage.2014.08.043 (2014).

32. Benveniste, H. *et al.* Anesthesia with Dexmedetomidine and Low-dose Isoflurane Increases Solute Transport via the Glymphatic Pathway in Rat Brain When Compared with High-dose Isoflurane. *Anesthesiology* 127, 976-988, doi:10.1097/ALN.0000000000001888 (2017).

33. Gass, N. *et al.* Brain network reorganization differs in response to stress in rats genetically predisposed to depression and stress-resilient rats. *Transl Psychiatry* 6, e970, doi:10.1038/tp.2016.233 (2016).

34. Baria, A. T. *et al.* BOLD temporal variability differentiates wakefulness from anesthesia-induced unconsciousness. *J Neurophysiol* 119, 834-848, doi:10.1152/jn.00714.2017 (2018).

35. Du, F. *et al.* Tightly coupled brain activity and cerebral ATP metabolic rate. *Proc Natl Acad Sci U S A* 105, 6409-6414, doi:10.1073/pnas.0710766105 (2008).

[Editors' note: further revisions were suggested prior to acceptance, as described below.]

The manuscript has been improved but there are some remaining issues that need to be addressed, as outlined below:In the Response to Reviewers, the authors present arguments that their anaesthesia protocol has minimal disruption on neurovascular coupling. Unfortunately, though, this does not come through in the discussion added to the paper itself, which focuses on respiration-LFP coupling. Given that the effects of anaesthesia on neurovascular coupling were very specifically raised by Reviewers 2 and 3 (and will be shared by many readers), please add a brief discussion along the lines of the response to Reviewer 2 Comment 3.

We apologize for missing the discussion of the anesthesia protocol in the main text. The following statements in our original response to Reviewer 2 Comment 3 have been added to the revised manuscript.

“Our rationale for choosing combined low-dose isoflurane and dexmedetomidine as our anesthesia protocol is as follows: Isoflurane is a vasodilator, which can attenuate the BOLD signal^75^, whereas the dexmedetomidine is a vasoconstrictor, which can counteract the vasodilatory effect from isoflurane^76^. This anesthesia protocol has been shown to minimize the confounding effects on the BOLD response and maintain the functional connectivity across both cortical and subcortical regions^77,78^.”